# Oxaliplatin Enhances the Apoptotic Effect of Mesenchymal Stem Cells, Delivering Soluble TRAIL in Chemoresistant Colorectal Cancer

**DOI:** 10.3390/ph16101448

**Published:** 2023-10-12

**Authors:** Adriana G Quiroz-Reyes, Paulina Delgado-González, José F. Islas, Adolfo Soto-Domínguez, Carlos A. González-Villarreal, Gerardo R. Padilla-Rivas, Elsa N. Garza-Treviño

**Affiliations:** 1Department of Biochemistry and Molecular Medicine, School of Medicine, Autonomous University of Nuevo Leon, Monterrey 81, Mexico; guadalupe.quirozrys@uanl.edu.mx (A.G.Q.-R.); paulina.delgadogn@uanl.edu.mx (P.D.-G.); jislas.me0117@uanl.edu.mx (J.F.I.); gerardo.padillarv@uanl.edu.mx (G.R.P.-R.); 2Department of Histology, School of Medicine, Autonomous University of Nuevo Leon, Monterrey 81, Mexico; adolfo.sotodmn@uanl.edu.mx; 3University of Monterrey, UDEM, San Pedro Garza Garcia 81, Mexico; carlosalberto.gonzalezv@udem.edu

**Keywords:** soluble TRAIL, mesenchymal stem cells, colorectal cancer

## Abstract

A key problem in colorectal cancer (CRC) is the development of resistance to current therapies due to the presence of cancer stem cells (CSC), which leads to poor prognosis. Tumor necrosis factor-related apoptosis-inducing ligand (TRAIL) is a protein that activates apoptosis in cancer cells through union with TRAIL death receptors. Cell therapies as delivery systems can produce soluble TRAIL (sTRAIL) and full-length TRAIL (flTRAIL), showing a high capacity to produce apoptosis in vitro and in vivo assays. However, the apoptotic activity of TRAIL as monotherapy had limitations, so it is important to explore other ways to enhance susceptibility to TRAIL. This study evaluated the cytotoxic and proapoptotic activity of soluble TRAIL overexpressed by mesenchymal stem cells (MSC) in an oxaliplatin-resistant CRC cell line. Bone marrow-MSC were lentiviral transduced for soluble TRAIL expression. DR5 death receptor expression was determined in Caco-2 and CMT-93 CRC cell lines. Sensitivity to first-line chemotherapies and recombinant TRAIL was evaluated by half-maximal inhibitory concentrations. Cytotoxic and proapoptotic activity of soluble TRAIL-MSC alone and combined with chemotherapy pre-treatment was evaluated using co-cultures. Caco-2 and CMT-93 cell lines expressed 59.08 ± 5.071 and 51.65 ± 11.99 of DR5 receptor and had IC_50_ of 534.15 ng/mL and 581.34 ng/mL for recombinant murine TRAIL (rmTRAIL), respectively. This finding was classified as moderate resistance to TRAIL. The Caco-2 cell line showed resistance to oxaliplatin and irinotecan. MSC successfully overexpressed soluble TRAIL and induced cancer cell death at a 1:6 ratio in co-culture. Oxaliplatin pre-treatment in the Caco-2 cell line increased the cell death percentage (50%) and apoptosis by sTRAIL. This finding was statistically different from the negative control (*p* < 0.05), and activity was even higher with the oxaliplatin–flTRAIL combination. Thus, oxaliplatin increases apoptotic activity induced by soluble TRAIL in a chemoresistant CRC cell line.

## 1. Introduction

Colorectal cancer (CRC) is the second cause of death-related cancer. It is associated with a late diagnosis and metastasis [1]. Chemoresistance is a major issue contributing to the poor prognosis in CRC treatment for advanced-stage tumors. Another problem is that current therapies are non-specific against cancer stem cells (CSC) [2]. This population of CSC has been associated with the maintenance of tumor growth and resistance to multiple therapies by activating DNA damage checkpoints, promoting epithelial to mesenchymal transition, over-expressing antiapoptotic factors, and ABC transporters, among others [3]. In addition, CSC has been identified by markers, such as CD24, CD44, and CD133, in multiple cancer types [4,5]. In CRC, our group identified overexpression of CD24 and CD44 in cells resistant to combination chemotherapy based on 5-fluorouracil and oxaliplatin (FOLFOX) [6].

TNF-related apoptosis-inducing ligand (TRAIL) has been used as an alternative treatment for cancer resistant to current chemotherapies. TRAIL activates extrinsic apoptosis by trimerization and activation of TRAIL death receptors DR4 and DR5, which leads to a Fas-associated death domain (FADD), which recruits caspase 8/10 to form the death-inducing signaling complex (DISC), leading to activation of a caspase cascade which triggers apoptotic changes and eventual cell death [7,8,9]. A limitation of using recombinant TRAIL therapies is its short life, which has reduced its application in clinical trials [10]. In addition, some cancer cells can develop resistance to recombinant TRAIL therapy by reducing TRAIL death receptors, overexpression of decoy receptors, antiapoptotic proteins such as c-FLIP and Bcl-2 family members, and decreasing proapoptotic factors [11]; however, the DR5 receptor can activate pro-survival signals in cancer cells [12]. 

Mesenchymal stem cells (MSC) are adult multipotent cells that have been developed as delivery vehicles for proapoptotic proteins, such as TRAIL. They can be found in different sources, such as adipose tissue, the umbilical cord, the placenta, and bone marrow [13]. This heterogeneous cell population can migrate into the tumor microenvironment by chemoattractants and continue releasing the protein into tumor cells [14,15,16]. Additionally, they can modulate the inflammatory process after injury [17]. In addition, different drugs, such as fluorouracil, doxorubicin, valproic acid, and paclitaxel, have shown a sensitization effect on cancer cells that increases TRAIL apoptosis, even in chemoresistant cell types [18,19,20,21,22]. Oxaliplatin is a DNA cross-linker that is applied as first-line chemotherapy in advanced-stage CRC [23]. This study aimed to show that pretreatment with oxaliplatin and MSC expressing soluble TRAIL in chemoresistant CRC cell lines increases the apoptotic effect of TRAIL and provides additional chemotherapy for resistant cancers.

## 2. Results

### 2.1. Lentiviral Transduced BM-MSC Successfully Expresses Soluble TRAIL

BM-MSC were genetically modified with lentivirus for soluble TRAIL expression, as described in our previous work [24]. In addition, we used lentivirus of full-length TRAIL as a control. MSC expressing soluble TRAIL (sTRAIL-MSC) and full-length TRAIL (flTRAIL-MSC) presented GFP-transduction marker signals by epifluorescence microscopy (Figure 1A). We analyzed protein expression to verify transgene integration into the host genome. The bands corresponding to TRAIL multimers of flTRAIL and a band of 24 KDa of monomeric sTRAIL (Figure 1B) were identified by Western blot. Quantification of sTRAIL levels in supernatants by ELISA showed a mean concentration of 327.01 ± 51.23 pg/mL. Moreover, TRAIL concentration in frozen supernatants (−80 °C) was quantified (255.03 ± 139.6 pg/mL). There was no statistically significant difference at 37 °C (*p* > 0.05, *t*-test) (Figure 1C). Thus, genetically modified MSCs express stable sTRAIL and flTRAIL.

### 2.2. Colorectal Cancer Cell Lines Present Chemoresistance to First-Line Treatment

We evaluated the sensitivity to conventional CRC treatment: 5-fluorouracil, oxaliplatin, and irinotecan by determining IC_50_ in cell lines_._ We then compared IC_50_ with a previously determined maximum peak plasma concentration of each drug and established the resistance sensitivity. Based on maximum peak plasma concentrations, CMT-93 was sensitive to the three chemotherapies, but Caco-2 cancer cells were resistant to oxaliplatin and irinotecan (Table 1). These results show that the colorectal cancer cell line Caco-2 is not sensitive to two chemotherapy drugs used as first-line treatment. 

### 2.3. Colorectal Cancer Cell Lines Are Susceptible to TRAIL

TRAIL DR5 receptor expression in colorectal cancer cell lines and susceptibility to recombinant TRAIL were analyzed to evaluate sensitivity to TRAIL. Caco-2 and CMT-93 cell lines expressed 59.08 ± 5.071 and 51.65 ± 11.99 of DR5 by immunofluorescence, respectively (Figure 2A,B). The DR5 receptor was uniform in the cell membrane. At 24 h, we found both cell lines had IC_50_ of 534.15 ng/mL and 581.34 ng/mL for rmTRAIL in Caco-2 and CMT-93, respectively. Thus, colorectal cancer cell lines were susceptible to rmTRAIL. 

### 2.4. MSC Expressing sTRAIL Induces Cell Death in Colorectal Cancer Cell Lines

Once TRAIL sensitivity of colorectal cancer cell lines was confirmed, we tested our expression system of sTRAIL by MSC. We included rmTRAIL and flTRAIL-MSC as positive controls. A cytotoxicity increase directly related to the number of MSCs added was found for both colorectal cancer cell lines. This finding shown in Figure 3 had a ratio of 1:6 with naïve-MSC, sTRAIL-MSC, and flTRAIL-MSC. In addition, we observed that naïve-MSC can induce cytotoxicity in the two tumor cell lines analyzed; however, naïve-MSC was more cytotoxic in CMT-93 at all ratios [cell death percentage (CDP) 20 to 75%] than Caco-2, which only had a significant cytotoxic effect at ratios of 1:3 and 1:6. The ratio 1:6 of sTRAIL-MSC induced an increase in cell death statistically different from the negative control (*p* < 0.05, Kruskal–Wallis, Dunn’s multiple comparisons) in the Caco-2 and CMT-93 cell lines (CDP 50 to 70) (Figure 3A,B). However, the CMT-93 cell line had a higher cell death percentage with naïve MSC treatment (CDP > 70) (*p* < 0.001, Kruskal–Wallis, Dunn’s multiple comparisons). In addition, there was no significant difference between sTRAIL treatments and flTRAIL and rmTRAIL controls. This finding implies that our sTRAIL delivery system can induce at least 50 of CDP in colorectal cancer cell lines. 

### 2.5. Combined Treatment of Oxaliplatin and MSC Expressing Soluble TRAIL Increases Apoptosis in a Chemoresistant Colorectal Cancer Cell Line

Previously, in this study, we observed chemoresistance to oxaliplatin in the Caco-2 cell line. However, we evaluated if this chemotherapy could increase the activity of TRAIL or as a cancer cell chemo sensitizer. To prove this, we applied a pretreatment of oxaliplatin at the maximum peak plasma concentration for 24 h and then treated cancer cells with sTRAIL-MSC 1:6 for another 24 h. As shown in Figure 4A,B, we found statistically significant differences in cytotoxicity induced by sTRAIL-MSC in both cell lines (CMT-93 and Caco-2) compared to the negative control. We observed an increase in cell death percentage with combinatory treatment of oxaliplatin and sTRAIL-MSC in the CMT-93 cell line (*p* < 0.01, Kruskal–Wallis, Dunn’s multiple comparisons); however, it was not statistically different from individual treatments for this cell line (Figure 4A). 

We also evaluated apoptosis in the Caco-2 cell line by caspase activation analysis, finding that combinatory treatment of oxaliplatin and sTRAIL-MSC presented a greater increase of caspase-3/7 activation (*p* < 0.0001, One-way ANOVA with Tukey’s multiple comparisons). In addition, an increase of sTRAIL-MSC activity when oxaliplatin pretreatment was applied was observed (*p* < 0.05 One-way ANOVA with Tukey’s multiple comparisons). This finding implies increased sensitivity of cells to TRAIL, and this activity was even higher than the combination of oxaliplatin with flTRAIL (Figure 4B).

### 2.6. Mesenchymal Stem Cell-Secreted sTRAIL Induces Cell Death in Colorectal Cancer Cell Lines Expressing Cancer Stem Cell Markers

Higher expression of CSC markers is present in chemoresistant cell lines [6]; thus, we evaluated the co-expression of CD24 and CD44 in cell lines. The Caco-2 cell line showed an expression of 56.44 ± 14.66, and CMT-93, an expression of 35.27 ± 19.28 of CD24^+^/CD44^+^ (Figure 5A,B). We then evaluated the activity of secreted sTRAIL by co-culture of Caco-2 cell line and sTRAIL-MSC with a 0.4 µm pore Transwell® filter (Costar, Corning, MA, USA). At 24 h of culture, we observed the detachment of cancer cells from the well surface. Remanent attached cells were stained by DAPI, and fluorescence intensity was quantified; there was a significant reduction of fluorescence signal when cells were exposed to sTRAIL alone or in combination with Oxaliplatin (*p* < 0.05, Kruskal–Wallis, Dunn’s multiple comparisons) (Figure 5C,D). Thus, delivered sTRAIL might induce cell death in colorectal cancer-expressing cancer stem cell markers.

## 3. Discussion

Colorectal cancer is still the second cause of cancer-related death, mainly because of the development of resistance to current therapies and metastasis [25,26]. In this study, we designed an expression system of soluble TRAIL by MSC to explore other strategies against colorectal cancer. Lentiviral-transduced MSC successfully expressed sTRAIL protein identified as a monomeric band by Western blot [27]. MSC constantly overexpressed TRAIL in supernatants at pg/mL concentrations, as reported by other groups [19,28,29]. In addition, sTRAIL-MSC expressed a functional protein that induced cytotoxicity against colorectal cancer cell lines, even when resistant to first-line chemotherapies. MSC cells alone can produce a cytotoxic effect on colorectal cancer cell lines, as in the CMT-93 cell line [30], and a poor effect in Caco-2. However, a pro-survival and regenerative effect is frequently observed with non-modified MSC, implying that its use is not recommended [31,32]. Other groups reported using sTRAIL delivered by MSC as an alternative cancer treatment since MSC presents tumor homing and immunomodulator characteristics that improve recombinant TRAIL monotherapy [28,33].

We evaluated TRAIL sensitivity by identifying TRAIL DR5 expression, finding that both cell lines expressed DR5 > 50%. It has been shown that TRAIL receptor expression is related to TRAIL sensitivity, in particular with DR5, as reduced DR5 levels are found in TRAIL-resistant cell lines, such as MSC [34]. In one study, DR4 suppression did not affect cell apoptosis. However, DR5 suppression massively blocked cancer cell apoptosis [12]. In colon cancer cell lines, both DR4 and DR5 receptors are required for maximum induction of apoptosis by recombinant TRAIL [35]. Nevertheless, colorectal cancer cell lines had moderate sensitivity to recombinant TRAIL since they require high protein concentrations for its IC_50_. TRAIL-resistant pancreatic cancer cell lines require concentrations above 500 ng/mL of recombinant TRAIL to reach IC_50_ [19]. Caco-2 was previously classified as resistant to TRAIL at 0.1 nM [36]. Other groups report disadvantages in using TRAIL as a recombinant protein because of its short life and fast elimination from the organism. Moreover, flTRAIL expressed in MSC could represent limitations in solid tumors [11,37].

Current cancer treatments focus on surgery, radiotherapy, and chemotherapeutic drugs. Colorectal cancer schemes for advanced cancer commonly include fluorouracil, capecitabine, and oxaliplatin [38]. Patients frequently develop resistance, resulting in treatment failure and poor outcomes [39]. Several groups have reported an additional use of this chemotherapy, as some of these agents can increase the susceptibility of cancer cells to TRAIL. It has been reported that fluorouracil, paclitaxel, and oxaliplatin can increase the sensitivity of colorectal cancer cells to TRAIL by reducing the expression of antiapoptotic molecules, such as c-FLIP, BCL-XL, NF-κβ, or increasing TRAIL death receptors [19,21,40]. In this study, oxaliplatin pretreatment increased the proapoptotic activity of sTRAIL-MSC, showing a chemo-sensitization effect on colorectal cancer cell lines. However, as a limitation of the study, it is necessary to determine the molecular mechanism of this TRAIL sensitization.

Another limitation of current treatments is the non-specific activity against CSC. This population of cells is related to chemoresistance development and tumor recurrence; current therapies are not specific against CSC [5,39]. We evaluated the expression rate of the CSC markers, CD24 and CD44, in colorectal cancer cell lines, finding high co-expression in both cell lines as in our previous study, which reported that expression of CSC markers in colorectal cancer cultures present resistance to the fluorouracil/oxaliplatin chemotherapeutic scheme [6]. Colorectal cancer cell lines with a higher population of CSC and their characteristics, as developers of resistant clones, also present resistance to oxaliplatin [41]. We observed that our treatment of sTRAIL-MSC induces apoptosis in this mixed population of colorectal cancer cells, highly expressing cancer stem cell markers.

We explored the activity of the sTRAIL protein without a cellular component, as it could be present in a solid tumor. We found a cellular detachment of cancer cells with both sTRAIL treatments or its combination with oxaliplatin. For relative quantification of cell death, we evaluated the fluorescent intensity of DAPI emitted by remanent adherent cells, showing a significant reduction of relative fluorescent units in tumoral cells treated with sTRAIL or its combination with oxaliplatin. Thus, it seems that even sTRAIL in monotherapy increases cell death in cells expressing cancer stem cell markers. 

## 4. Materials and Methods

### 4.1. MSC Isolation and Characterization

We isolated a primary culture and propagated it in bone marrow-mesenchymal stem cells (BM-MSC) from mice. BM-MSC isolation and characterization were developed as described in our previous work [24]. Briefly: BM-MSC were obtained from 6 to 8-week-old Balb/c mice sacrificed in a CO_2_ chamber. Cells were deposited into a 25 cm^2^ culture flask (Corning Inc, One Riverfront Plaza, Corning, New York, NY 14831, USA) with DMEM/F12-GlutaMAX (Gibco, Life Technologies, Grand Island, NY, USA) supplemented with 10% fetal bovine serum (FBS) (Gibco, Life Technologies, Grand Island, NY, USA), gentamicin (100 µg/mL) (Gibco, Life Technologies, Grand Island, NY, USA), amphotericin B (2.5 µg/mL) (Gibco, Life Technologies, Grand Island, NY, USA), and glutamine (2 nm) (Gibco, Life Technologies, Grand Island, NY, USA). The following immunohistochemistry markers, CD105, CD90, and CD34, were used to identify MSC with the primary monoclonal antibodies anti-CD90, anti-CD105, and anti-CD34 (United States Biologicals, Salem, MA, USA), as the ISCT specifies [42]. In addition, the Mouse Mesenchymal Stem Cell Functional Identification Kit (R&D Systems, Inc. Minneapolis, MN, USA) was used for multipotency evaluation of BM-MSC to differentiate osteoblasts and adipocytes. 

### 4.2. Lentiviral Transduction of BM-MSC

The lentivirus was previously designed by our group [43] with the vector building platform by Cyagen (Santa Clara, CA, USA) for expression of soluble TRAIL (sTRAIL) and full-length TRAIL (flTRAIL): pLV[Exp]-EGFP/Neo-EF1A > {sMurTRAIL}, pLV[Exp]-EGFP/Neo-EF1A > {flMurTRAIL}. Transduction was performed following the method by Yuan et al. [44]. Culture flasks were seeded with 5 x10^5^ BM-MSC and incubated for 16 h with 5% CO_2_ at 37 °C. Protamine sulfate, 5 µg/mL (Sigma-Aldrich, Merck, St. Louis, MO, USA), was added after 16 h. All cultures were infected with lentivirus at 2 MOI (multiplicity of infection). Next, cells were incubated for an additional 48 h under the same conditions, and then geneticin (400 µg/mL) (Gibco, Life Technologies, Grand Island, NY, USA) was added. Transduction efficiency was evaluated by fluorescent count using epifluorescence microscopy (Nokia, Eclipse 50i).

### 4.3. Transgene Expression Validation and Quantification

TRAIL expression was shown by Western blot. After 48 h of culture, 9 ± 1 mL of culture medium was collected from flasks with transduced cells of sTRAIL and flTRAIL genes. Next, the culture medium was concentrated by ultrafiltration and centrifugation in an Amicon® Ultra-15 10K Centrifugal Filter Device column (Merck-Millipore, Burlington, MA, USA) at 5000 RPM for 40 min at room temperature. Proteins were quantified with Bradford reactive Bradford Quick Start (Bio-Rad, Hercules, CA, USA). Polyacrylamide gel electrophoresis (12%) was run with these preparations. Gels were loaded with 40 µL (approx. 100 µg/mL of total protein) and run for 10 min at 80 V and then for 1 h at 100 V. Proteins were transferred to a PVDF membrane (Bio-Rad, Hercules, CA, USA) and blocked overnight with silk milk 5% (Svelty, Nestlé, Mexico City, Mexico) in TBS in agitation (20 RPM) at 4 °C. A PVDF membrane was then incubated with mouse primary anti-TRAIL (ab2435) antibodies (1:200) and anti-β actin (ab1801) (1:1000) antibodies obtained from rabbit (Abcam, Burlingame, CA, USA). Anti-rabbit secondary polyclonal antibody conjugated with HRP (hose-radish peroxidase, 1:10,000) was also used. The Luminol Clarity ECL Kit (Bio-Rad, Hercules, CA, USA) was used to reveal the Western blot membrane. Soluble TRAIL levels were quantified by ELISA using the TRAIL ELISA kit from Abcam (ab253210). sTRAIL-MSC 48 h culture supernatants were collected and centrifuged at 2000× *g* for 10 min. After that, ELISA was developed following the manufacturer’s instructions. A Cytation 3 (BIOTEK, Winooski, VT, USA) reader was used for TRAIL determination.

### 4.4. Colorectal Cancer Cell Lines

The adenocarcinoma cell lines Caco-2 (ATCC® HTB-37) and CMT-93 (ATCC® CCL-223^TM^) were used for assays. Both cell lines were cultured in DMEM 1X-GlutaMAX (Gibco, Life Technologies, Grand Island, NY, USA) supplemented with 10% FBS (Gibco, Life Technologies, Grand Island, NY, USA), gentamicin (100 µg/mL) (Gibco, Life Technologies, Grand Island, NY, USA), amphotericin B (2.5 µg/mL) (Gibco, Life Technologies, Grand Island, NY, USA), and glutamine (2 nm) (Gibco, Life Technologies, Grand Island, NY, USA).

### 4.5. Susceptibility to TRAIL by Evaluating DR5 Receptor Expression

The TRAIL death receptor (DR5) was analyzed by immunofluorescence. Cell lines were cultured in Lab-Tek® microchambers (Nunc®, Thermo Scientific, Waltham, Massachusetts, USA) and, after 24 h, were fixed with methanol-acetone (1:1 *v/v*) and washed three times with Tris-Buffered saline Tween 20 (TBST). Slides were then incubated with primary antibodies anti-DR5 (ab8416, Abcam, Burlingame, CA, USA) (1:1000) and incubated at 4 °C for 12 h. The slides were washed again three times with TBST. The secondary antibody Alexa Fluor® 488 (ab150113, Abcam, Burlingame, CA, USA) (1:1000) was added and incubated for 1 h at 25 °C. All antibodies were diluted in phosphate saline buffer (PBS). VECTASHIELD® 25 µL mixed with 4′,6′-diamino-2-phenilindol (DAPI) (Vector Laboratories, Inc. 6737 Mowry Ave Newark, CA 94560 USA) was added as a mounting medium. Slides were observed with a ZEISS Imager.A2 epifluorescence microscope (Carls ZEISS, Alemania). Five pictures were taken per slide with an AxioCam MRc coupled to the microscope. Images were analyzed with ZEISS ZEN Blue and Image J software. The expression percentage was calculated with relative luminescence units as the (positive Red-Green/DAPI signal) × 100.

### 4.6. Determination of the Half-Maximal Inhibitory Concentration (IC_50_)

We determined the IC_50_ of chemotherapy and recombinant TRAIL. Colorectal cancer cell lines were cultured in 96-well plates at a density of 2 × 10^3^ cells per well and incubated for 24 h. Then the chemotherapeutics, 5-fluorouracil (5FU) (Fresenius Kabi, Lake Zurich, IL, USA), oxaliplatin (Tiboquir®, Ulsa Tech, Av. Dr. Roberto Michel no. 2546Col. Parque Industrial el Alamo, C.P.44490 Guadalajara, Jalisco, Mexico), and irinotecan (Colizactive, Glenmark Pharmaceuticals, Bombay, India) were added at a concentration range of 0–80 µg/mL and incubated for 48 h. Recombinant murine TRAIL (rmTRAIL) (PeproTech, 315-19, ThermoFisher Scientific, Waltham, MA, USA) was applied at different concentrations (0.8 ng/mL–12.5 µg/mL) and incubated for 24 h. ATP luminescence was determined by CellTiter-Glo (Promega, Madison, WI, USA) to evaluate the cell death percentage. CellTiter-Glo reactive (100 µL) was added to each well, agitated for 2 min at 300–500 rpm, and incubated for 10 min at room temperature. Luminescence was quantified in a Cytation 3 plate reader (BioTek, Winooski, VT, USA). Cell death percentage was calculated with the formula: [1-(treatment mean luminescence/control mean luminescence)] × 100. IC_50_ was calculated with linear regression.

### 4.7. Cytotoxicity and Apoptotic Effect of MSC Expressing TRAIL by Co-Culture Assay

Colorectal cancer cell lines were cultured in 96-well plates at a density of 2 × 10^3^ cells per well and incubated for 24 h with or without oxaliplatin pretreatment at a maximum peak plasma concentration (2.9 µg/mL) [6] (Tiboquir®, Ulsa Tech, Av. Dr. Roberto Michel no. 2546Col. Parque Industrial el Alamo, C.P.44490 Guadalajara, Jalisco, Mexico). After that time, MSC expressing sTRAIL, flTRAIL, or naïve were co-cultured at 1:1, 1:3, or 1:6 ratio and incubated for 24 h to determine the best cytotoxic MSC ratio. Recombinant murine TRAIL protein at IC_50_ was used as a positive control. For cytotoxic analysis, 100 µL of CellTiter-Glo reactive was added to each well, agitated for 2 min at 300–500 rpm, and incubated for 10 min at room temperature. Luminescence was quantified in a Cytation 3 plate reader (BioTek). Cell death percentage was calculated with the formula: [1 − (mean luminescence–effector mean luminescence/control mean luminescence)] × 100. Evaluation of activated caspases luminescence signal was associated with apoptosis. The colorectal cancer cell line Caco-2 was cultured in 96-well plates at a density of 2 × 10^3^ cells per well and incubated 24 h with oxaliplatin (2.9 µg/mL) (Tiboquir®, Ulsa Tech, Av. Dr. Roberto Michel no. 2546Col. Parque Industrial el Alamo, C.P.44490 Guadalajara, Jalisco, Mexico). Co-culture with 12 × 10^3^ MSC expressing sTRAIL or flTRAIL was incubated for another 24 h. Apoptosis was analyzed with a Caspase-Glo 3/7® (Promega, Madison, WI, USA) kit. The 100 µL reagent was added to each well, agitated for 30 s, and incubated for 2 h at room temperature. Luminescence was quantified in a Cytation 3 plate reader (BioTek). 

### 4.8. Expression of CSC Markers in Colorectal Cancer Cell Lines and Cell Death Evaluation by Fluorescence

The CSC markers CD44 and CD24 were analyzed by immunofluorescence. Cell lines were cultured in Lab-Tek® microchambers (Nunc®, Thermo Scientific). After 24 h, they were fixed with methanol-acetone (1:1 *v/v*) and washed three times with Tris-Buffered saline Tween 20 (TBST). Slides were then incubated with the primary antibodies, anti-CD44 conjugated with PE/Cy7 (1:100), and anti-CD24 conjugated with FITC (1:100). All antibodies were diluted in PBS and incubated at 4 °C for 12 h. The slides were washed again three times with TBST. The secondary antibody Alexa Fluor® 488 (1:1000 in PBS) was added and incubated for 1 h at 25 °C. As a mounting medium, 25 µL of VECTASHIELD® mixed with DAPI (Vector Laboratories, Inc. 6737 Mowry Ave Newark, CA 94560, USA) was added. Slides were observed with a ZEISS Imager.A2 epifluorescence microscope (Carls ZEISS), taking five pictures per slide with an AxioCam MRc coupled to the microscope. Images were analyzed with ZEISS ZEN Blue and Image J software. The expression percentage was calculated with relative luminescence units as: (positive Red-Green/DAPI signal) × 100. We evaluated cell death by fluorescence. The Caco-2 cell line was cultured in 6-well plates at a concentration of 5 × 10^4^ cells per well above a coverslip. After cell attachment to the coverslip, we applied pretreatment with oxaliplatin (2.9 µg/mL) (Tiboquir®, Ulsa Tech, Av. Dr. Roberto Michel no. 2546Col. Parque Industrial el Alamo, C.P.44490 Guadalajara, Jalisco, Mexico) and incubated for 24 h. Using 0.4 µm Transwell® (Corning Inc, One Riverfront Plaza, Corning, New York, NY 14831, USA), we co-culture 3 × 10^5^ MSC expressing sTRAIL and incubate for 24 h. After that, we removed the coverslips with cells and added 25 µL of VECTASHIELD® mixed DAPI (Vector Laboratories, Inc. 6737 Mowry Ave Newark, CA 94560, USA). Slides were observed with a ZEISS Imager.A2 epifluorescence microscope (Carls ZEISS), taking five pictures per slide with an AxioCam MRc coupled to the microscope. Images were analyzed with ZEISS ZEN Blue and Image J software.

### 4.9. Statistical Analysis

The *t*-test, the Mann–Whitney U, one-factor ANOVA, and the Kruskal–Wallis tests were used for group comparisons. The results were interpreted with SPSS(version 22.0., IBM TechXchange Community, 1 New Orchard Road Armonk, New York, NY 10504-1722, USA), GraphPad Prisma(version 9.0.0, GraphPad Software, 225 Franklin Street. Fl. 26 Boston, MA 02110, USA) (significance *p* < 0.05). All experiments were performed twice in triplicate.

## 5. Conclusions

According to the results, MSCs are a promising strategy as a delivery system for antiapoptotic proteins such as sTRAIL and flTRAIL, which can induce cytotoxicity and apoptosis in CRC cell lines. We found a ratio of 1:3 and 1:6 with naïve-MSC, sTRAIL-MSC, and flTRAIL-MSC-induced cell death and apoptosis against tumor cells (Caco-2 and CMT-93) that had moderate resistance to recombinant TRAIL which was statistically different from the negative control. We also demonstrated increases in the antitumoral activity of MSC expressing sTRAIL at a ratio of 1:6 with oxaliplatin pretreatment. Our treatment with sTRAIL-MSC as monotherapy induces apoptosis in colorectal cancer cells expressing cancer stem cell markers. This treatment is effective even in a chemoresistant CRC cell line.

## Figures and Tables

**Figure 1 pharmaceuticals-16-01448-f001:**
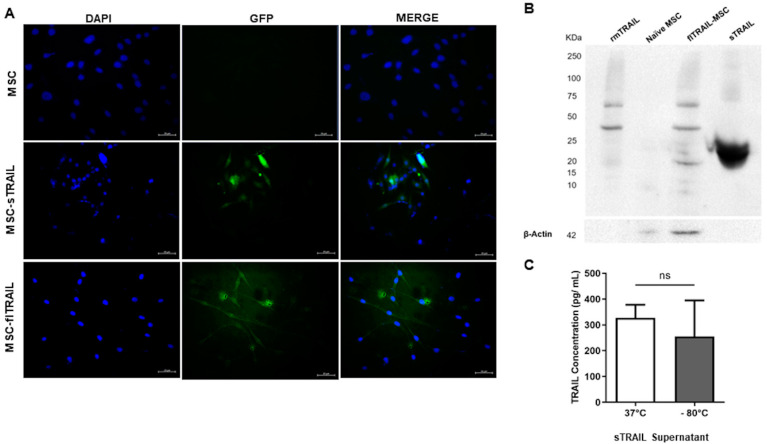
**Mesenchymal stem cell genetically modified to express TRAIL transgene.** (**A**) Transduced MSC-TRAIL express GFP (200×). (**B**) Western blot of TRAIL bands. flTRAIL presented multiple bands corresponding to TRAIL multimeters, while sTRAIL only presented a monomeric band at 25 kDa. Controls of recombinant murine TRAIL and naïve MSC were included. (**C**) Quantification of TRAIL at 48 h in MSC-sTRAIL supernatants (37 °C and −80 °C). (37 °C and −80 °C). *p* > 0.05, *t*-test. Data of mean and standard error. MSC: Mesenchymal stem cells; sTRAIL: soluble TRAIL; flTRAIL: full-length TRAIL; rmTRAIL: recombinant murine TRAIL. GFP: Green fluorescent protein.

**Figure 2 pharmaceuticals-16-01448-f002:**
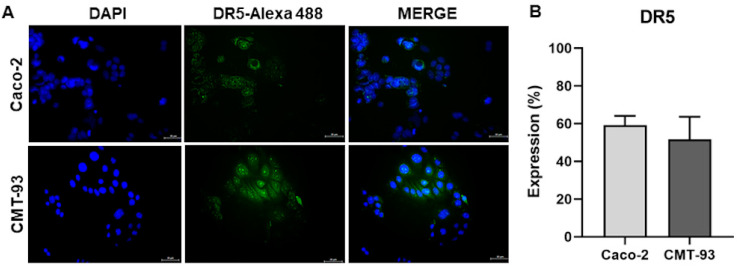
**TRAIL death receptor expression in colorectal cancer cell lines.** (**A**) DR5 fluorescent expression in Caco-2 and CMT-93 cell lines (200×). Secondary antibody labeled with fluorophore Alexa 488. (**B**) Quantification of DR5 expression percentage in Caco-2 and CMT-93 (mean and standard deviation).

**Figure 3 pharmaceuticals-16-01448-f003:**
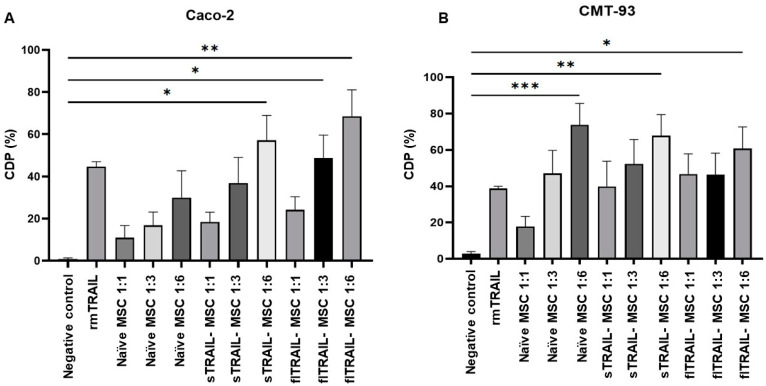
**MSC expressing TRAIL activity in colorectal cancer cell lines.** MSC-sTRAIL at a 1:6 ratio increased cell death percentage in (**A**) Caco-2 and (**B**) CMT-93 cell lines. *p* < 0.05 *, *p* < 0.01 **, *p* < 0.001 ***. (Kruskal–Wallis, Dunn’s multiple comparisons (mean and standard error). Controls of naïve MSC and fl-TRAIL were included at the same ratio. CDP: Cell death percentage. MSC: Mesenchymal stem cells; sTRAIL: soluble TRAIL; flTRAIL: full-length TRAIL; rmTRAIL: recombinant murine TRAIL. GFP: Green fluorescent protein.

**Figure 4 pharmaceuticals-16-01448-f004:**
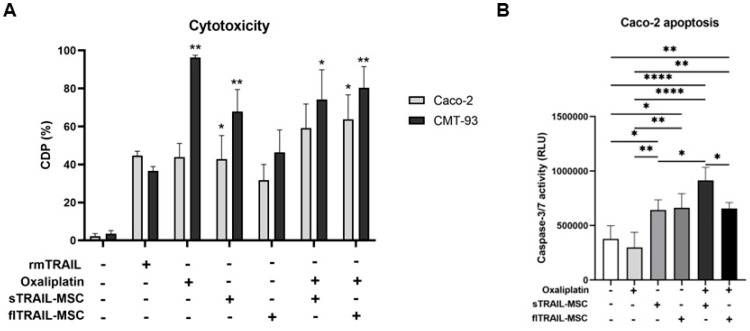
Oxaliplatin increases the antitumoral activity of MSC expressing soluble TRAIL in chemoresistant colorectal cancer cell lines. (**A**) Cell death percentage of TRAIL and oxaliplatin treatments. (**B**) Apoptotic activity of TRAIL and oxaliplatin treatments. *p* < 0.05 *, *p* < 0.01 **, *p* < 0.0001 ****. (Kruskal–Wallis, Dunn’s multiple comparisons (mean and standard error). One-factor Anova, Tukey’s multiple comparisons (mean and standard deviation). CDP: Cell death percentage. MSC: Mesenchymal stem cells; sTRAIL: soluble TRAIL; flTRAIL: full-length TRAIL; rmTRAIL: recombinant murine TRAIL. GFP: Green fluorescent protein.

**Figure 5 pharmaceuticals-16-01448-f005:**
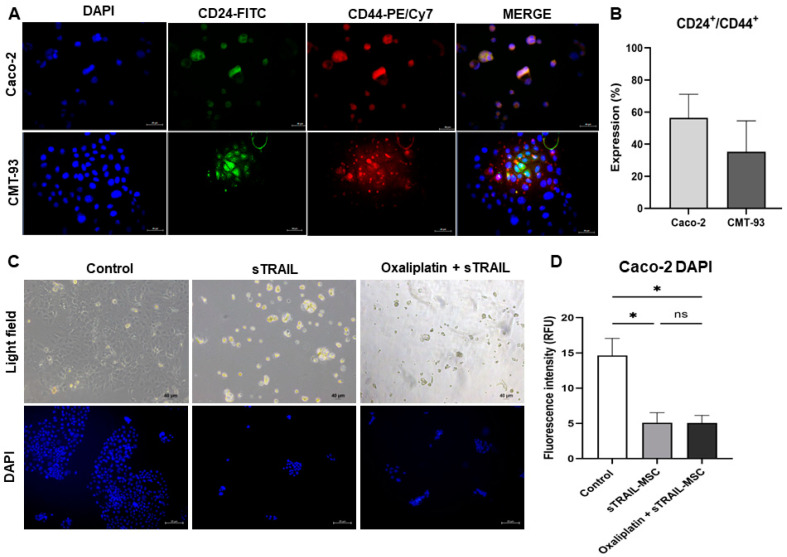
**Colorectal cancer cell lines highly express cancer stem cell markers and are susceptible to sTRAIL cell death.** (**A**) CD24 and CD44 expression by colorectal cancer stem cell lines by fluorescence (200×). (**B**) Quantification of co-expression percentage of CD24/CD44 in colorectal cancer cell lines Caco-2 and CMT-93 (mean and standard deviation). (**C**) Light field and fluorescence micrographics of Caco-2 cell line with sTRAL treatment or combined with oxaliplatin (100×). (**D**) Quantification of DAPI fluorescent intensity of Caco-2 cells after treatments. *p* < 0.05 *, ns: non-significant (Kruskal–Wallis, Dunn’s multiple comparisons (mean and standard error). MSC: Mesenchymal stem cells; sTRAIL: soluble TRAIL; RFU: relative fluorescent units.

**Table 1 pharmaceuticals-16-01448-t001:** Colorectal cancer cell lines susceptibility to first-line chemotherapy.

	5-Fluorouracil		Oxaliplatin		Irinotecan		Reference
	IC_50_ (µg/mL)	Max Plasma (µg/mL)	IC_50_ (µg/mL)	Max Plasma (µg/mL)	IC_50_ (µg/mL)	Max Plasma (µg/mL)	
CACO-2	1.66	10	4.0	2.9	2.14	1.97	[6]
CMT-93	0.096	10	0.15	2.9	1.33	1.97	

## Data Availability

Data is contained within the article.

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
