# Peer review of "Oxaliplatin Enhances the Apoptotic Effect of Mesenchymal Stem Cells, Delivering Soluble TRAIL in Chemoresistant Colorectal Cancer"

_pharmaceuticals, 2023, doi:10.3390/ph16101448_

Round 1

Reviewer 1 Report

In this study, the authors evaluated the cytotoxic and proapoptotic activity of soluble TRAIL overexpressed by MSCs on an oxaliplatin-resistant CRC cell line. The observation in this study is interesting, but there are several issues that should be addressed.

Lines 86-88, the authors stated that “TRAIL concentration in frozen supernatants (-80°C) was quantified, 86 and there was no statistically significant difference (255.03 ± 241.87 pg/ mL) (p>0.05, t-test) 87 (Figure 1C).” However, there is no comparison in Figure 1C. Only one column in this figure, who compares with whom?

In Figure 3, it showed MSC-sTRAIL/flTRAIL at ratios of 1:1, 1:3, 1:6, what is the reason for choosing these ratios? And how to determine this ratio? Please descript it more detail in Methods.

In this study, MSCs were from mice, while colorectal cancer cell line Caco-2 was from human. Whether there is an impact of cells from different species on the experimental results? In addition, the results from normal colon epithelial cells is also encouraged.

Although many interesting observations in this study, it seems there is a lack of further mechanism exploration.

Minors:

The period in the end of the Title should be removed.

Line 82, “gen” should be corrected.

Figure 4A, “citotoxity” should be corrected.

Author Response

-Lines 86-88, the authors stated that “TRAIL concentration in frozen supernatants (-80°C) was quantified, 86 and there was no statistically significant difference (255.03 ± 241.87 pg/ mL) (p>0.05, t-test) 87 (Figure 1C).” However, there is no comparison in Figure 1C. Only one column in this figure, who compares with whom?

Response: We appreciate the comments to improve our work. Originally, the graph of sTRAIL concentration at - 80°C was not shown in Figure 1 due to it was not statistically different, but we decided to add and correct it to improve the understanding of the information.

-In Figure 3, it showed MSC-sTRAIL/flTRAIL at ratios of 1:1, 1:3, 1:6, what is the reason for choosing these ratios? And how to determine this ratio? Please descript it more detail in Methods.

Response: The ratios of MSC co-culture with tumoral cells is based in other research from different research group (Fakiruddin, K.S.; Lim, M.N.; Nordin, N.; Rosli, R.; Zakaria, Z.; Abdullah, S. Targeting of CD133+ Cancer Stem Cells by Mesenchymal Stem Cell Expressing TRAIL Reveals a Prospective Role of Apoptotic Gene Regulation in Non-Small Cell Lung Cancer. Cancers 2019, 11, 1261. https://doi.org/10.3390/cancers11091261). As it was our first approach, we needed to select the best concentration of MSC expressing soluble TRAIL that produced a statistically significant increase in cell death. This clarification was included in the main text.

In this study, MSCs were from mice, while colorectal cancer cell line Caco-2 was from human. Whether there is an impact of cells from different species on the experimental results? In addition, the results from normal colon epithelial cells is also encouraged.

Response: We selected MSC from mice because of lentiviral vector is for murine-soluble TRAIL. Our delivery system previously showed anti-tumoral activity against mice lymphoma, thus we would like to know if it also presented anti-tumoral activity against human cancer cells. We performed a protein blast of human and mouse TRAIL amino acid sequences and found a percentage of identity of 65.31 %. Moreover, DR5 receptor has the same structure in human and mouse. MSC presents the advantage of immunotolerance, which is an important characteristic for being used as delivery vectors. In this first approach, we only evaluated tumoral cells, however, in future works, we would like to test this treatment on healthy colon cells.

Although many interesting observations in this study, it seems there is a lack of further mechanism exploration.

Response: As you mention, in this work we have not yet characterized the mechanism by which there is greater sensitization to TRAIL using Oxaliplatin. We hope in future studies we can be clearer about the mechanism. Therefore, we added this limitation in the manuscript in the discussion section.

 Minors:

The period in the end of the Title should be removed.

Line 82, “gen” should be corrected.

Figure 4A, “citotoxity” should be corrected.

Response: From the minor reviews, the period at the end of the Title was removed. Line 82, “gen” was substituted by “transgene”. The word “Cytotoxicity” on Figure 4A was corrected.

Reviewer 2 Report

Colorectal cancer (CRC) is a major clinical challenge due to the presence of cancer stem cells (CSCS) and the lack of effective treatment options. IIn this study, the cytotoxic and pro-apoptotic effects of soluble TRAIL overexpressed by mesenchymal stem cells on oxaliplatin resistant colorectal cancer cell lines were evaluated. The research content conforms to the journal field. However, in its present form, the manuscript has some shortcomings that hinder its recommendation for publication. The authors need to answer the following questions:

1.        There are issues with the English writing that are beyond the scope of research. There are also missing words in sentences, and fragmented sentences. Suggest to get a native English writer to improve the manuscript.

2.        In Figure 1B, the western blot experimental results are not satisfactory, and it is suggested to improve them.

3.        In Figure 1C, the results appear to lack a control group.

4.        It would be better to mark scale bars in all immunofluorescence stained pictures.

5.        Most recent studies about cancer and mesenchymal stem cells related mechanism, such as Biomaterials Translational, 2023, 4(2):  67-84; Biomaterials Translational, 2020, 1(1): 33-45; Journal of Orthopaedic Translation, 2023, 39: 88-99; Journal of Orthopaedic Translation, 2023, 39: 63-73; etc., are recommended to be cited in introduction and discussion section to support your results and conclusion.

 There are issues with the English writing that are beyond the scope of research. There are also missing words in sentences, and fragmented sentences. Suggest to get a native English writer to improve the manuscript.

Author Response

  1. There are issues with the English writing that are beyond the scope of research. There are also missing words in sentences, and fragmented sentences. Suggest to get a native English writer to improve the manuscript.

Response: Thank you for taking the time to review our manuscript. We appreciate the comments to improve our work. The manuscript was reviewed by a medical certificate writer.

  1. In Figure 1B, the western blot experimental results are not satisfactory, and it is suggested to improve them.

Response: We improve the definition of western blot from Figure 1B reducing background noise, and showing clear sharp bands.

  1. In Figure 1C, the results appear to lack a control group.

Response: In Figure 1C we only determined TRAIL concentrations on TRAIL supernatants, but we performed a standard curve. For that reason, we did not include another control group.

  1. It would be better to mark scale bars in all immunofluorescence stained pictures.

Response: We added all scale bars in the immunofluorescence results.

  1. Most recent studies about cancer and mesenchymal stem cells related mechanism, such as Biomaterials Translational, 2023, 4(2):  67-84; Biomaterials Translational, 2020, 1(1): 33-45; Journal of Orthopaedic Translation, 2023, 39: 88-99; Journal of Orthopaedic Translation, 2023, 39: 63-73; etc., are recommended to be cited in introduction and discussion section to support your results and conclusion.

Response: We appreciate the suggestions and added some sentences in the introduction and discussion section from the texts that you recommended.

Reviewer 3 Report

The manuscript title “ Oxaliplatin enhances apoptotic effect of Mesenchymal Stem Cells delivering Soluble TRAIL on chemoresistant colorectal cancer” is an interesting and may be acceptable with some corrections as per below:

Comments:

1.      In abstract author does not added the results/data which is required to understand the research article in beginning.

2.      Line 53, elaborate and expand the “FOLFOX” for better understanding.

3.      Line 58, elaborate and expand the “FADD” for better understanding.

4.      Line 86 and 87, the sTRAIL mean concentration 328.3 ± 90.13 through ELISA, and the value 255.03 ± 241.87 pg/ mL,  in these the SD value is too much  high, not acceptable.

5.      The conclusion is not written well only three lines?

NA

Author Response

1.In abstract author does not added the results/data which is required to understand the research article in beginning.

Response: Thank you for your manuscript suggestions. We added some results in the abstract. 

  1. Line 53, elaborate and expand the “FOLFOX” for better understanding.

Response: We re-write the sentence for better understanding.

  1. Line 58, elaborate and expand the “FADD” for better understanding.

Response: We re-write this part of the text.

  1. Line 86 and 87, the sTRAIL mean concentration 328.3 ± 90.13 through ELISA, and the value 255.03 ± 241.87 pg/ mL, in these the SD value is too much high, not acceptable.

Response: We correct the results on sentences in lines 86 and 87 (ELISA levels of sTRAIL at 37ºC and after freeze -80ºC) and added in the Figure 1C.

  1. The conclusion is not written well only three lines?

Response: We re-write the conclusion section.

Reviewer 4 Report

Manuscript has a good title. English language has good quality. figures have acceptable quality. There are some modifications that are essential to be exerted in the manuscript.

1. About page 75, line 150-151

Why you let the concentration of oxaliplatin reach at the maximum peak plasma concentration?

2. About line 156-157 in page 5

Why do you think that it was not statistically different from individual treatments in this section of manuscript?

3. Abput line 158-160 in page 5

Why you only evaluated caspase activity? Was it not better to find receptors that involve in the process of apoptosis in mentioned

cancer cells and after that, you connect caspase activity, remarked receptors and sTRAIL-MSC in combined with oxaliplatin with together and display all mentioned connection in the form of a simple figure?

4. Line 181-182 in page 6

Why the detachment of cancer cells from the well surface was occurred?

Do you think remanent attached cells were appropriate cases for DAPI staining? If yes, why?

5. About figures

+ Please make the scales of all figure more visible

+ Please determine the important details in each figure

6. About line 196 in page 7, section "3. Discussion"

Please insert a simple figure in which the summary of your present manuscript is demistrated

7. Please check and adjust the "Reference list" based on the regulations of reference list of journal. (Titles, doi, the name of journal and ... )

Author Response

  1. About page 75, line 150-151. Why you let the concentration of oxaliplatin reach at the maximum peak plasma concentration?

Response: Thank you for taking the time to review our manuscript. We appreciate the comments to improve our work. To answer your first question, commonly colorectal cancer patients present resistance to this concentration of oxaliplatin. Thus, we select the maximum peak plasma concentration of oxaliplatin to translate our results to clinical patients. This concentration was determined before for our research group for cells isolated from colorectal cancer tumor biopsies (Garza‑Treviño, E.; Martínez‑Rodríguez, H.; Delgado‑González, P.; Solís‑Coronado, O.; Ortíz‑Lopez, R.; Soto‑Domínguez, A.; Treviño, V.; Padilla‑Rivas, G.; Islas‑Cisneros, J.; Quiroz‑Reyes, A.; et al. Chemosensitivity analysis and study of gene resistance on tumors and cancer stem cell isolates from patients with colorectal cancer. Mol. Med. Rep. 2021, 24, 1–13, doi:10.3892/mmr.2021.12360).

  1. About line 156-157 in page 5. Why do you think that it was not statistically different from individual treatments in this section of manuscript?

Response: We observed that CMT-93 cell line was very sensitive to the treatments of oxaliplatin and MSC expressing sTRAIL, thus the individual or combinatory treatments generate high percentages of cell death. Moreover, at comparing the combination with individual treatments, the gap in cell death percentage is too short and for that reason, the differences were not statistically significant. 

  1. About line 158-160 in page 5 Why you only evaluated caspase activity? Was it not better to find receptors that involve in the process of apoptosis in mentioned cancer cells and after that, you connect caspase activity, remarked receptors and sTRAIL-MSC in combined with oxaliplatin with together and display all mentioned connection in the form of a simple figure?

Response: TRAIL extrinsic pathway apoptosis is mediated by the activation of caspases. In this study, we only evaluated caspase activity as a first approach to this mechanism of cell death. As shown in the results, we confirm that the cell death process was due to apoptosis in colorectal cancer cell lines. This was the first step in the study; however, we are considering the determination of TRAIL receptors for posterior research.

  1. Line 181-182 in page 6 Why the detachment of cancer cells from the well surface was occurred? Do you think remanent attached cells were appropriate cases for DAPI staining? If yes, why?

Response: Caco-2 colorectal cancer cell line present in vitro an adherent morphology. Its form of growing is around the stem cell. When cells die, they detach from the cell surface as lost anchor proteins by the apoptosis process. Thus, we observed that adding sTRAIL and Oxaliplatin plus sTRAIL treatment induced cell death as evaluated by cytotoxic and apoptotic assays. However, as in other cancer models, there is a high variability of response to treatments, and some cells will survive. For DAPI stain, we made washes with PBS to elute non-adherent cells and stain the remaining. DAPI signal from remanent cells shows us the viability that remains after treatment and a relative cell death quantification. Another group also used DAPI quantification for determining cell viability (Baer, P.C.; Koch, B.; Freitag, J.; Schubert, R.; Geiger, H. No cytotoxic and inflammatory effects of empagliflozin and dapagliflozin on primary renal proximal tubular epithelial cells under diabetic conditions in vitro. Int. J. Mol. Sci. 2020, 21, doi:10.3390/ijms21020391).

  1. About figures

+ Please make the scales of all figure more visible

+ Please determine the important details in each figure

Response: We increase the scales in figures, add the important details in figure descriptions, and check the reference list to include all the information.

6. About line 196 in page 7, section "3. Discussion". Please insert a simple figure in which the summary of your present manuscript is demonstrated

 Response: As you require a simple figure to summarize the demonstrated data, we designed a graphical abstract.

7. Please check and adjust the "Reference list" based on the regulations of the reference list of journal. (Titles, doi, the name of journal and ... )

Response: We verify the reference list and adjust it by the journal.

Round 2

Reviewer 1 Report

The authors have made efforts to revise the manuscript. Their revision is appropriate and the response to my comments is satisfactory.

Reviewer 4 Report

No more comment.